# A Scoping Review of the Skeletal Effects of Naringenin

**DOI:** 10.3390/nu14224851

**Published:** 2022-11-16

**Authors:** Muhamed Lahtif Nor Muhamad, Sophia Ogechi Ekeuku, Sok-Kuan Wong, Kok-Yong Chin

**Affiliations:** Department of Pharmacology, Faculty of Medicine, Universiti Kebangsaan Malaysia, Cheras, Kuala Lumpur 56000, Malaysia

**Keywords:** naringenin, osteoblasts, osteoclasts, osteocytes, osteoporosis

## Abstract

Background: Osteoporosis is caused by the deterioration of bone density and microstructure, resulting in increased fracture risk. It transpires due to an imbalanced skeletal remodelling process favouring bone resorption. Various natural compounds can positively influence the skeletal remodelling process, of which naringenin is a candidate. Naringenin is an anti-inflammatory and antioxidant compound found in citrus fruits and grapefruit. This systematic review aims to present an overview of the available evidence on the skeletal protective effects of naringenin. Method: A systematic literature search was conducted using the PubMed and Scopus databases in August 2022. Original research articles using cells, animals, or humans to investigate the bone protective effects of naringenin were included. Results: Sixteen eligible articles were included in this review. The existing evidence suggested that naringenin enhanced osteoblastogenesis and bone formation through BMP-2/p38MAPK/Runx2/Osx, SDF-1/CXCR4, and PI3K/Akt/*c*-Fos/*c*-Jun/AP-1 signalling pathways. Naringenin also inhibited osteoclastogenesis and bone resorption by inhibiting inflammation and the RANKL pathway. Conclusions: Naringenin enhances bone formation while suppressing bone resorption, thus achieving its skeletal protective effects. It could be incorporated into the diet through fruit intake or supplements to prevent bone loss.

## 1. Introduction

Osteoporosis is a degenerative skeletal condition characterised by reduced bone mass and microstructural deterioration, which subsequently lead to decreased bone strength and an increased fragility fracture risk [1]. Osteoporosis is asymptomatic until it presents as low-trauma fractures of the hip, spinal, proximal humerus, pelvis, and/or wrist [2]. Osteoporosis is more prominent in postmenopausal women because estrogen insufficiency accelerates bone loss [3]. The use of antiresorptive (e.g., bisphosphonates, denosumab, and selective estrogen receptor modulators) and anabolic medications (e.g., teriparatide and abaloparatide) can improve bone mineral density (BMD) and reduce the fracture risk of patients with osteoporosis [4,5]. However, they come with various side effects [6,7].

Bone loss occurs when the rate of osteoblastic bone formation is lower than the rate of osteoclastic bone resorption [8]. Various factors could influence the bone turnover process. Inflammation, known to promote bone resorption, is a risk factor for osteoporosis [9]. Proinflammatory cytokines stimulate the expression of receptor activators of nuclear factor-B (RANK) and its functional ligand (RANKL), along with macrophage colony-stimulating factor (M-CSF), which enhance osteoclast formation and function [8]. Furthermore, modifications in redox systems have been linked to the pathogenesis of osteoporosis. Reactive oxygen species (ROS) inhibit osteoblast formation, stimulate apoptosis in osteoblasts and osteocytes, and encourage the formation of osteoclasts [10], all of which result in bone loss and osteoporosis.

Apart from calcium and vitamin D routinely used in osteoporosis prevention [11], dietary antioxidants and anti-inflammatory compounds may slow the progression of osteoporosis [10,12]. Naringenin (Figure 1) is a flavanone present in citrus fruits, grapes, and tomato skin [13]. Naringenin has been investigated for its antioxidant [14] and anti-inflammatory properties [15]. Previous research found that naringenin suppressed nuclear factor-kappa B (NF-kB) p65 activity and expression in streptozotocin (STZ)-induced diabetic mice [16] and carrageenan-induced paw oedema in rats [17]. In STZ and nicotinamide-induced diabetic rats, naringenin significantly increased the activities of pancreatic enzymatic antioxidants, plasma non-enzymatic antioxidant levels and decreased pancreatic tissue malondialdehyde levels [18]. Meanwhile, another report indicated that naringenin lowered lipid peroxidation and enhanced the activity of antioxidant enzymes, such as superoxide dismutase, catalase, glutathione-s-transferase, glutathione peroxidase and reduced glutathione in the liver of STZ-induced diabetic mice [19]. The studies mentioned above point to the potential of naringenin as an antioxidant and anti-inflammatory agent, which could help to suppress bone loss.

Considering the effects of inflammation and antioxidants in the pathogenesis of osteoporosis, naringenin may be positioned as a functional food component in preventing bone loss. The objective of this review is to encapsulate the protective effects of naringenin on bone as evidenced by currently available studies. Furthermore, naringenin may also exert specific mechanisms to enhance bone health, which would be discussed in the current review.

## 2. Materials and Methods

### 2.1. Literature Review

This systematic review followed the Preferred Reporting Items for Systematic Reviews and Meta-Analyses (PRISMA) guidelines for scoping review (Appendix A). To discern studies on the potential benefits of naringenin on bone, a systematic literature search was conducted using the PubMed and Scopus databases in August 2021. The keywords used in the search were (1) naringenin AND (2) (bone OR osteoporosis OR osteoblasts OR osteoclasts OR osteocytes).

### 2.2. Article Selection

Articles with the following features were included: (1) original research articles investigating the skeletal effects of naringenin; (2) studies conducted using cell cultures, animal models, or human subjects. Articles with the following characteristics were rejected: (1) conference abstracts, reviews, letters or commentary, editorial and opinion articles lacking original data; (2) studies not using pure naringenin; (3) articles written in a language other than English. The search was executed by two authors (MLNM and SOE), using both databases and the keywords listed. The inclusion of an article was based on consensus by both authors. If no consensus could be obtained, the opinions of other authors were sought after (SKW and KYC) to determine the outcome of the article.

### 2.3. Data Extraction

The list of articles was organised using Mendeley (Elsevier, Amsterdam, The Netherlands). Identification of duplicated items was performed using Mendeley and manually. The authors’ names, publication year, study design, dose, treatment period, outcomes, and study limitations were all extracted using a standardised table by two authors (MLNM and SOE).

## 3. Results

### 3.1. Article Selection

The literature search resulted in 210 unique articles, which were from PubMed and Scopus. Following the removal of duplicates (*n* = 51), 159 articles were screened. One hundred and forty-three articles were excluded because they did not fit the inclusion criteria (not original article = 63, out of scope = 60, not written in English = 4, raw extract/mixture/formulation = 16). Eventually, the review included 16 articles that met all the criteria (Table 1). The selection process is depicted in Figure 2.

### 3.2. Study Characteristics

The studies selected for this review were published from 2005 to 2021. There were eleven in vitro studies included in this review. The studies were either osteoblastogenesis/osteogenesis or osteoclastogenesis. For osteoblastogenesis/osteogenesis studies, cells used were rat bone marrow stroma cells (rBMSCs), mouse calvarial osteoblasts, bone marrow-derived mesenchymal stem cells (BMSCs), rat/mouse calvarial osteoblasts, MC3T3-E1, human foetal osteoblasts (hOB), primary osteoblast cells (pOB) and human periodontal ligament stem cells (hPDLSCs) [21,22,23,26,27,28,29]. The osteogenic effect of naringenin was evaluated through dexamethasone-induced osteogenesis [21,26].

For osteoclastogenesis studies, cells used were rabbit osteoclasts, human primary osteoclasts precursor cells, co-culture of T-cells with bone marrow macrophages (BMMs), bone marrow monocytes (BMM), mouse MBMMφ and RAW 264.7 cell line [20,22,24,25,30]. Macrophage colony-stimulating factor (M-CSF) and receptor activator of NF-kB ligand (RANKL) were used to induce the conversion of osteoclast precursor cells into mature osteoclasts to investigate the effects of naringenin on osteoclast differentiation [20,22,24,25,30]. Doses of naringenin used in the in vitro studies ranged from 0 to 800 µM. The treatment duration for osteoclastogenesis differentiation was about 2–6 days, while osteogenesis differentiation lasted 14–16 days.

Meanwhile, eight in vivo studies involved using rats (Wistar, Sprague Dawley, Y59) and mice (Balb/cByJ, *C*-57/BL6, and ICR) [23,25,27,29,31,32,33,34]. In animal studies, naringenin doses ranged from 0.005–25 mg/kg, 3–10 mg/mL, and 0.09–0.72% weight of diet [23,25,27,29,31,32,33,34]. The disease models used in these studies were ovariectomy (OVX)/oestrogen deficiency, retinoic acid-induced bone loss, titanium (Ti) particle-induced osteolysis, and soft diet-induced periodontal hypofunction. In these studies, the bone structure was determined using micro-computed tomography and histomorphometry. The bone remodelling process was investigated using circulating bone remodelling markers. The treatment duration for in vivo studies was between 2–6 weeks. There was no finding from human studies on this topic. The effects of naringenin on bone health are summed up in Table 1.

### 3.3. Findings from In Vitro Studies

Osteoblasts are mesenchymal cells that support bone formation by producing a bone matrix and subsequently mineralising it [35]. Naringenin was found to inhibit etoposide- and tumour necrosis factor-alpha (TNF-α)-induced osteoblast cell apoptosis in murine primary osteoblastic cells [27]. Naringenin improved osteoblast differentiation, mineralization, and osteogenic function in cultured rat calvarial osteoblasts [22,23,26,27], rBMSCs [21], and hPDLSCs [28] through the bone morphogenetic protein-2 (BMP-2)/p38 mitogen-activated protein kinase (p38 MAPK)/runt-related transcription factor2 (Runx2)/Osterix (Osx)/alkaline phosphatase (ALP) signalling pathway. Enhanced expressions of C-X-C chemokine receptor type 4 (CXCR4) and stromal cell-derived factor 1 (SDF-1a) levels were observed in naringenin-treated BMSCs [26] and hPDLSCs [28], suggesting that naringenin regulated osteogenic differentiation through the SDF-1/CXCR4 signalling pathway. Naringenin also exerted BMP-dependent osteogenic effects via the phosphoinositide 3-kinase (PI3K), protein kinase B (Akt), *c*-Fos/*c*-Jun and activator protein 1 (AP-1) dependent signalling pathways [27]. Another study also found that naringenin increased OPG/RANKL ratio based on mRNA and protein expression from osteoblasts [22]. This alteration can potentially alter osteoclast formation. However, no changes in osteocalcin (OCN) were observed in MG-63 cell lines treated with naringenin. The possible molecular mechanisms of naringenin’s effect on osteoblasts are summarized in Figure 3.

Osteoclasts are multinucleated hematopoietic cells that serve as the main bone-resorbing cells and play an important role in bone remodelling [36]. Naringenin significantly inhibited osteoclastogenesis and secretion of interleukin (IL)-1α, IL-23 as well monocyte chemoattractant protein-1 in pre-osteoclast cultures. A significant decrease in helical peptide 620–633 release indicating bone resorption activity by naringenin, and an increase in tumour necrosis factor-α (TNF-α), IL-8, and IL-4 [24] levels were observed in human pre-osteoclasts, but no changes were found on IL-6, IL-1b, IL-17, and osteoprotegerin (OPG) levels [20]. Besides, naringenin also was found to inhibit osteoclast formation and bone resorption activity, indicated by reduced numbers of osteoclasts, bone resorption pits, and area, as well as markers of osteoclast maturation, such as tartrate-resistant acid phosphatase (TRAP) and cathepsin-K [22,24,25]. Naringenin was proven to minimize the number and size of F-actin rings, lower the expressions of cathepsin K, *c*-Fos, dendritic cell-specific transmembrane protein (DC-STAMP), nuclear factor of activated T-cell 1 (NFATc1), tartrate-resistant acid phosphatase (TRAP) and vacuolar type proton-translocating ATPase (V-ATPase d2) of osteoclasts at mRNA and protein levels [24,25]. Furthermore, naringenin has been shown to reduce the RANKL-induced p38 phosphorylation signalling in pre-osteoclasts and cause no changes in gene expression of nuclear factor-κB (NF-κB), extracellular signal-regulated kinase (ERK), *c*-Jun *N*-terminal kinase (JNK), nuclear factor kappa light polypeptide gene enhancer B-cells inhibitor alpha (IκBα) and phosphorylated nuclear factor kappa light polypeptide gene enhancer B-cells inhibitor alpha (*p*-IκBα) [25]. The possible mechanism of actions of naringenin on osteoclasts are summarised in Figure 4.

### 3.4. Findings from In Vivo Studies

Postmenopausal bone loss due to oestrogen deficiency is caused by a high bone remodelling phenomenon that results in an imbalance between bone formation and resorption [37]. Gera et al., (2022) [29] reported that naringenin supplementation (20 mg/kg for 60 days) in female Wistar rats with OVX-induced osteoporosis increased ALP and decreased acid phosphatase in the serum. They also reported improved cortical and trabecular bone architecture following naringenin supplementation. Kaczmarczyk-Sedlak et al., 2016 [31] reported that naringenin supplementation (50 mg/kg for 4 weeks) in female Wistar rats with OVX-induced osteoporosis increased the width of trabecular in the epiphysis, decreased organic substances in the tibia, periosteal transverse growth of the diaphysis as well lower the ratio of the transverse cross-section area of the marrow cavity/diaphysis. However, the study showed no skeletal biomechanical and mineral (calcium and phosphate) changes probably because of the short duration of supplementation. The supplementation also did not alter the transverse cross-sectional area of cortical bone in the diaphysis and cartilage width compared to the negative group.

Swarnkar et al., (2012) [23] discovered that naringenin (10 mg/kg for 6 weeks) treatment in Balb/cByJ mice mitigated OVX-induced trabecular bone changes by increasing the mineralized nodule, mineral apposition rate and bone formation rate/bone surface (BFR/BS) compared to negative group. This effect was probably attributed to improved osteoblastogenesis indicated by the mRNA gene expression of runt-related transcription factor 2 (Runx-2) and type I collagen. In a preventive model, the same group revealed a lack of changes in trabecular microstructure at the distal femoral epiphysis and tibial proximal metaphysis. In the therapeutic model (5 mg/kg/day of naringenin (i.p.) for 6 weeks), they reported decreased trabecular bone pattern factor but no changes in trabecular microstructure in the distal femoral epiphysis and tibial proximal metaphysis with supplementation. Wu et al., (2008) [27] also claimed that treating OVX ICR mice with naringenin (10 mg/mL every 2 days for 4 weeks) could enhance BMD, bone mineral content, ALP activity, and BMP-2 expression.

Retinoic acid-induced bone loss in rats has been used to evaluate the skeletal impacts of multiple substances on the skeletal system [38]. According to Oršolić et al., (2014) [32], the administration of naringenin (100 mg/kg for 14 days) in Y59 female rats with retinoic acid-induced bone loss resulted in increased calcium and phosphorus content in the femur, higher BMD in the proximal and distal femur and improved femur length. They attribute the protection to the improvement of the redox status of the rats, but the redox parameters, such as glutathione and malondialdehyde, were measured in the liver and kidney. Oršolić et al., (2022) [34] also reported that the administration of naringenin (100 mg/kg for 14 days) in Y59 female rats with retinoic acid-induced bone loss resulted in decreased serum β-CTx, IL-1β, IL-6, TNFα, and chemokine ligand 5 (CCL5/RANTES). They also reported decreased MDA and increased GSH and CAT in the ovary and kidney. However, they reported a lack of changes in circulating bone formation marker (OCN) level, circulating, bone calcium, and phosphorus levels, densitometry, and cortical geometry in the supplemented group. A short duration of treatment might be responsible for the lack of skeletal effects of the treatment.

Osteolysis is the progressive degeneration of periprosthetic bony tissue, which appears on serial radiographs as progressive radiolucent lines and/or cavitation at the implant-bone or cement-bone interface. Osteolysis can progress to aseptic loosening and implant failure if not treated promptly [39]. Based on the study performed by Wang et al., (2014) [25], naringenin supplementation (10 and 25 mg/kg for 2 weeks) reduced TRAP-positive multinucleated osteoclasts and lowered the number of pores and the percentage porosity in the calvarial region of interest in *C*-57/BL6 mice with Ti-particle induce osteolysis. Moreover, an increase in trabecular bone volume was reported compared to the negative group.

Periodontal hypofunction can occur as a consequence of disassociation with an opposing tooth in certain malocclusions, such as open bites and ectopic teeth [40]. Wood (2005) [33] showed that naringenin supplementation (0.09%, 0.18%, 0.36%, and 0.72% for 42 days) reduced physiological molar crestal alveolar bone (CAB)-cemento-enamel junction (CEJ) distance in buccal maxilla and mandible as well as in lingua maxilla and mandible in Sprague Dawley rats with soft diet-induced periodontal hypofunction compared to the unsupplemented group.

## 4. Discussion

The currently available evidence shows that naringenin exerts stimulatory effects on osteoblasts through MAPK, PI3K/Akt, and CXCR4/SDF-1 pathways. Osteoblasts are responsible for osteogenesis by synthesising and mineralising organic bone matrix (osteoid) during skeleton construction and bone remodelling [41]. The MAPK cascades regulate Runx2 phosphorylation and transcription, which promote osteoblast differentiation. MAPK pathways and their components, JNK, ERK, and p38, which enforce osteoblastogenesis and establish the non-canonical BMP-2 signal transduction pathways [42,43,44,45]. Naringenin stimulates osteogenic gene activation, indicating that it has a stimulatory effect on osteogenic differentiation [21,22]. Activation of the PI3K/Akt signalling pathway also promotes osteoblast proliferation, differentiation, and bone formation activity [46]. Naringenin has been reported to stimulate BMP-2-dependent osteoblastogenesis through the activation of the PI3K/Akt signalling pathway [27]. Activation of the CXCR4/SDF-1 signalling pathway is critical in early osteoblastogenesis and its suppression leads to lower bone formation and mineralisation [47,48].

Osteoclasts are bone resorption cells originating from hematopoietic lineage cells [49]. Bone resorption is vital in bone remodelling, but excessive resorption can result in pathological bone loss. Osteoclast differentiation and activation are governed by various hormones and cytokines. The cytokines RANKL and M-CSF, in particular, are required for osteoclastic differentiation [50]. To promote osteoclast differentiation, preservation and bone resorption, the M-CSF binds to the colony-stimulating factor 1 receptor, whilst RANKL binds to the RANK receptor [51,52]. TRAF factors such as TRAF 6 are recruited by RANK-RANKL binding [53], leading to the activation of NF-kB, Akt, and MAPKs (ERK/p38/JNK) pathways. Furthermore, the RANKL signalling stimulates *c*-Fos and then NFATc1, a major switch that plays a role in controlling osteoclast terminal differentiation [54,55]. Naringenin was reported to reduce M-CSF and RANKL-induced expression of critical markers of osteoclast differentiation markers such as cathepsin K, *c*-Fos, and NFATc1 [24,25].

The biological properties of naringenin suggest a broad range of clinical applications. Naringenin decreased CAB-CEJ distance in buccal maxilla and mandible as well as in lingua maxilla and mandible, indicating that naringenin supplementation protects against alveolar bone loss in rats with induced periodontal disease [33]. Supplementation of naringenin also improved bone mineral microstructure, mineral, and biomechanical strength [23,27,31,32]. However, the bone loss models that have been used to test the effects of naringenin have been limited to OVX and retinol-induced models. Thus, results from other models, such as testosterone deficiency and glucocorticoid models, are indispensable before it is tested on patients with other causes of osteoporosis. Following joint replacement, the abrasive particles initiated by the prosthesis are primarily responsible for osteolysis [56]. Naringenin supplementation prevented Ti-particle-induced osteolysis, implying that it may be preferable for treating periprosthetic osteolysis [25].

Pharmacokinetics and safety issues of naringenin should be considered before it is used clinically. From the pharmacokinetic aspects, naringenin has very low in vivo bioavailability due to its hydrophobic nature, which limits its practical use. It has a short half-life and is easily converted to its crystalline form, and therefore it is poorly absorbed by the digestive system [57,58,59,60]. Previous researchers developed a variety of methods to improve naringenin absorption and low bioavailability, including particle size reduction, complexation with cyclodextrins [61], salt formation, solid dispersions [62], surfactant usage, nanoparticles, nanocarriers [63], and self-emulsifying drug delivery system, as well as prodrug formation [64]. Nanotechnology proved to be an efficient way to improve the bioavailability of naringenin by multiple delivery routes to enhance its effectiveness in the treatment of cancer, inflammation, diabetes, liver, brain, and ocular diseases mostly through numerous in vitro and in vivo methods [65]. Meanwhile, Rodríguez-Fragoso et al., (2011) [66] found out that naringenin inhibits some drug-metabolizing cytochrome P450 enzymes, including CYP3A4 and CYP1A2, potentially leading to drug-drug interactions in the intestine and liver, where phytochemical concentrations are higher. Modification in cytochrome P450 and other enzymatic activity may influence the outcome of drugs that go through extensive first-pass metabolism. An acute toxicity study using Wistar rats reported the lethal dose (LD_50_) value of naringenin to be 340 mg/kg body weight [67]. Using body surface ratio conversation [68], the human equivalent dose is 64 mg/kg.

The term “naringenin” was searched for on https://clinicaltrials.gov/ (accessed on 31 August 2022) and the search revealed thirteen registered clinical trials on naringenin. The trials investigate the effects of naringenin on healthy subjects (NCT02627547, NCT04867655, NCT05073523, NCT02380144), hepatitis virus/HCV infection/chronic HCV/Hepatitis C (NCT01091077), energy expenditure/safety issues/glucose metabolism (NCT04697355), safety issues/pharmacokinetics (NCT03582553), subjective cognitive decline (NCT04744922), cardiovascular disease risk factors (NCT00539916), intestinal disease (NCT03032861), metabolic syndrome/vascular compliance/predisposition to cardiovascular disease (NCT04731987), pharmacokinetics of new curcumin formulations/safety of new curcumin formulations (NCT01982734) and cardiovascular risk factor/type-2 diabetes mellitus/insulin sensitivity/metabolic syndrome (NCT03527277). Seven of these clinical trials have been completed, four are still recruiting, one with an unknown status, and lastly one trial is still active but not recruiting. However, no attempt has been made to conduct a human clinical trial to evaluate the impact of naringenin on skeletal diseases. Since pure naringenin has only been studied in limited clinical trials, more research on free drug and naringenin-loaded nanosystems in humans is warranted. Further exploration into the interactions of these nanoformulations with the human body is required before they can be translated into pharmaceuticals and nutraceutical supplements [69].

This review also has several limitations. We only include articles written in English in this review, which potentially excludes studies published in other languages. We did not exclude studies based on their quality because the number of studies is limited. Nevertheless, the current review provides an overview of the skeletal effects of naringenin and prospects of its clinical application as a functional food component to protect bone health.

## 5. Conclusions

Preclinical findings demonstrate that naringenin protects the skeleton by suppressing osteoclastogenesis and bone resorption while enhancing osteoblastogenesis and bone formation. Human clinical trials to justify naringenin’s skeletal effects are lacking. Hence, comprehensive clinical studies should be performed to validate naringenin’s skeletal properties and reveal the safety of this flavanone compound.

## Figures and Tables

**Figure 1 nutrients-14-04851-f001:**
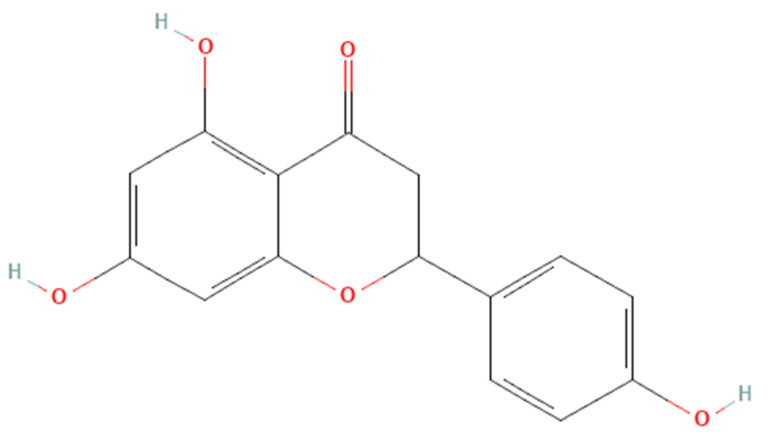
The molecular structure of naringenin.

**Figure 2 nutrients-14-04851-f002:**
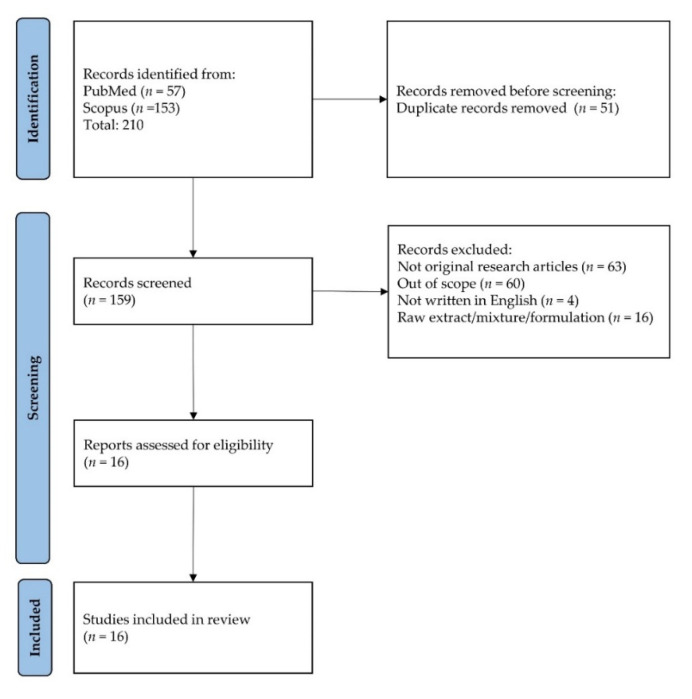
Flowchart showing selection of articles.

**Figure 3 nutrients-14-04851-f003:**
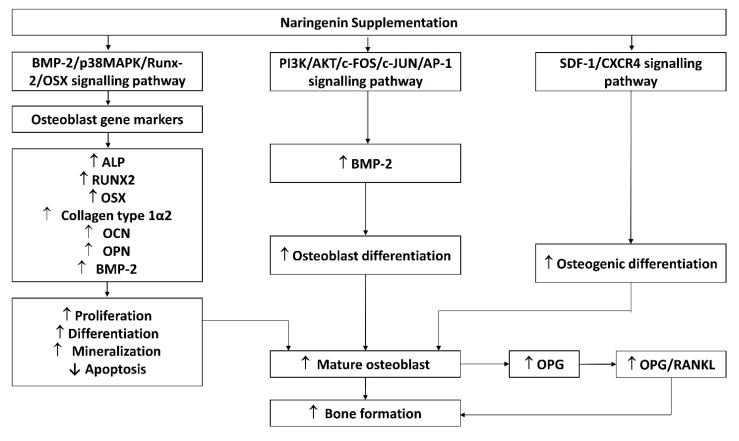
Possible mechanism of naringenin effect on osteoblasts. Abbreviations: ALP, alkaline phosphatase; IL-1β, OPG, osteoprotegerin; OSX, osterix; RANK, receptor activator of nuclear factor-kappa B ligand; BMP-2; p38MAPK, phosphoinositide 38 mitogen-activated protein kinase; Runx2, runt-related transcription factor2; OCN, osteocalcin; OPN, osteopontin; P13K, phosphoinositide 3-kinase (PI3K); AKT, protein kinase B; *c*-FOS; *c*-JUN; AP-1, activator protein 1; SDF-1, stromal cell-derived factor 1; CXCR4, C-X-C chemokine receptor type 4.

**Figure 4 nutrients-14-04851-f004:**
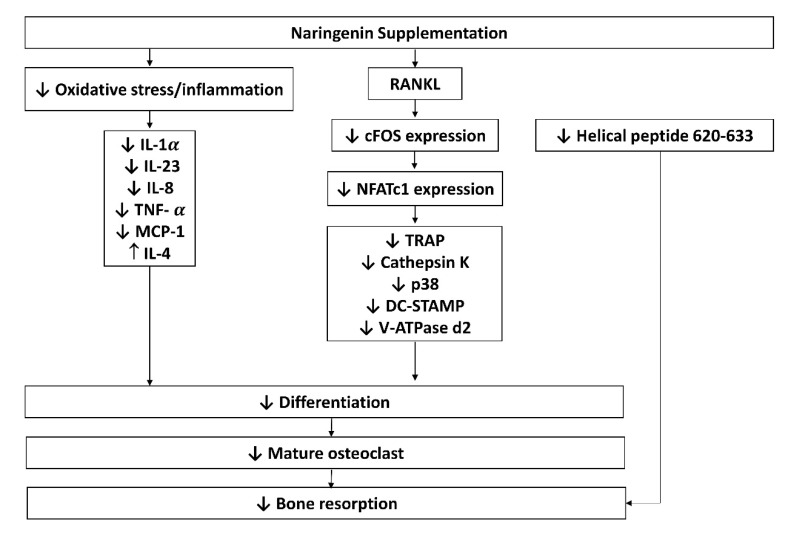
Possible mechanism of actions of naringenin on osteoclasts. Abbreviations: IL-1 α, -23, -8, -4, interleukin-1 alpha, -23, -8, -4; NFATc1, nuclear factor of activated T- cells; OPG, osteoprotegerin; OSX, osterix; RANKL, receptor activator of nuclear factor-kappa B; ligand; TNF-α, tumour necrosis factor-alpha; TRAP, tartrate-resistant acid phosphatase; p38, phosphoinositide 38; DC-STAMP, dendritic cell-specific transmembrane protein; V-ATPase d2, vacuolar-type proton-translocating ATPase; MCP-1, monocyte chemoattractant protein-1.

**Table 1 nutrients-14-04851-t001:** Effects of naringenin on bone health.

**Cell Culture Studies**		**Major Findings (Changes vs. Negative Control)**			**Conclusion**
**Reference**	**Study Design**	**Indices Increased**	**Indices Decreased**	**Indices Unchanged**	
[20]	Cell: Human primary osteoclast precursor cells Induction: RANKL- and M-CSF-induced osteoclastogenesis Treatment: 36.7, 91.8, and 183.6 μM of naringenin (2 μL per well) for 2 and 6 days. Control: Negative: No treatment Positive: No	IL-8 and TNF-α secretion	Osteoclastogenesis IL-1α, IL-23 and monocyte chemoattractant protein-1 secretion Helical peptide 620–633 release	IL-6, IL-1b, IL-17, and OPG secretion	Naringenin inhibits osteoclast formation and bone resorption.
[21]	Cell: rBMSCs from femur and tibia of Wistar rats Induction: Dexamethasone-induced osteogenesis Treatment: 10 μM of naringenin or 8-prenylnaringenin for 16 days Control Negative: No treatment Positive: n.a	ALP activity Osteocalcin level Calcium level BMP-2, OPG, Runx-2 and Osx expression p38 MAPK expression at 12 and 24 h Effects of prenylnaringenin > naringenin			Both prenylnaringenin and naringenin promote osteoblast differentiation and mineralization. Effects of prenylnaringenin are better than naringenin.
[22]	Cell: Calvarial osteoblasts from newborn Wistar rats and osteoclasts cells from femur and tibia of rabbit Induction: n.a Treatment: 10 μM of naringenin and 8-prenylnaringenin for 3, 6, 9, and 12 days Control: Negative: No treatment Positive: n.a	ALP activity at all time points Osteocalcin expression (day 3–6 and day 6–9) Calcium content (day 6) numbers and areas of mineralized nodules (day 12) Runx-2, Osx, BMP-2 and Col1α2 expression OPG expression, OPG/RANKL ratio Apoptotic osteoclast at 24 h	Osteoclast number, number of pits, and total area of pits TRAP-positive multinuclear cells and expression of TRAP and cathepsin-K	RANKL expression	Both prenylnaringenin and naringenin suppress osteoclast formation and survival and promote osteoblast differentiation. Effects of prenylnaringenin are better than naringenin.
[23]	Cell: Mouse calvarial osteoblasts from Balb/cByJ mice Induction: n.a Treatment: Naringenin or naringenin-6-*C*-glucoside (0.001,0.01 and 0.1, 1, 10, 25, and 50 μM) for 21 days Control: Negative: No treatment Positive: 17β-oestradiol (1 nM) for 21 days	ALP activity (25 and 50 μM for naringenin, 1–100 nM for naringenin-6-*C*-glucoside) Nodule formation (50 μM for naringenin, 100 nM for naringenin-6-*C*-glucoside)			Both naringenin or naringenin-6-*C*-glucoside promote osteoblast differentiation and bone formation. Naringenin-6-*C*-glucoside is more potent than naringenin.
[24]	Cell: co-culture of T-cells from male BALB/c mice and BMMs Induction: M-CSF- and RANKL-induced osteoclastogenesis Treatment: Naringenin (0–800 μM) for 1, 2, and 10 days. Control: Negative: no treatment Positive: n. a	IL-4 release by T-cells	TRAP-positive multinucleated osteoclast Size and number of F-actin rings mRNA and protein expression of cathepsin K, *c*-Fos, DC-STAMP, NFATc1, TRAP, and V-ATPase d2		The anti-osteoclastogenesis effects of naringenin are exerted through IL-4 release by T-cells.
[25]	Cell: BMMs from femur and tibia of *C*-57/BL6 mice and RAW 264.7 cells. Induction: M-CSF- and RANKL-induced osteoclastogenesis. Treatment: Naringenin (0–800 μM) for 2 days. Control: Negative: no treatment Positive: n. a		Size and number of F-actin ring Resorption area RANKL-induced cathepsin K, *c*-Fos, DC-STAMP, NFATc1, TRAP, and V-ATPase d2 RANKL-induced phosphorylation of p38 signalling	NF-κB, ERK, JNK, IκBα, and *p*-IκBα expression	Naringenin inhibits osteoclast formation through suppression of p38 signaling.
[26]	Cell: BMSCs from femur and tibia of Sprague Dawley rats Induction: dexamethasone-induced osteogenesis Treatment: 734.6 μM of naringenin for 14 days Control: Negative: No treatment Positive: n. a	mRNA and protein expression of ALP, Runx-2, CXCR4, and SDF-1a		Cell viability and proliferation rates	Naringenin stimulates bone formation via upregulation of the SDF-1a through the SDF-1/CXCR4 signaling pathway
[27]	Cell: MC3T3-E1, hOB and pOB cell line Induction: n.a Treatment: 0.3, 1, 3, and 10 μM of naringenin for 1–7 days Control: Negative: no treatment Positive: n.a	ALP activity Osteocalcin and osteopontin expression BMP-2 via PI3K and Akt-dependent signaling proliferation of MC3T3-E1	BMP-3 expression Etoposide- and TNF-α-induced cell apoptosis		The osteogenic effects of naringin are exerted through upregulation of BMP-2 expression via the PI3K, Akt, *c*-Fos/*c*-Jun and AP-1-dependent signaling pathway
[28]	Cell: hPDLSCs Induction: osteogenic differentiation Treatment: 0.1, 1 and 10 mM of naringenin for 0–72 h Control: Negative: no treatment Positive: n.a	ALP activity (day 3 and 7) SDF-1 mRNA expression SDF-1 and CXCR4 protein expression			Naringenin increases the osteogenic potential of hPDLSCs.
[29]	Cell: MG-63 cell lines Induction: n.a. Treatment: 100 μL of 0.15–10 μg/mL naringenin nanosuspension for 48 h Control: Negative: no treatment Positive: n.a			OCN protein expression	Naringenin nanosuspension may have pro-osteogenic effects.
[30]	Cell: mouse MBMMφ and RAW 264.7 cells Induction: M-CSF- and RANKL-induced osteoclastogenesis Treatment: 2.5, 5, and 10 μg/mL naringenin for 72 h Control: Negative: no treatment Positive: n.a			RANKL-induced osteoclastogenesis and resorption area	Naringenin suppresses osteoclast formation and resorption activity.
**Animal Studies**		**Major Findings (Changes vs. Negative Control)**			
**Researchers (Year)**	**Study Design**	**Indices Increased**	**Indices Decreased**	**Indices Unchanged**	
[31]	Animal: 28 Female Wistar rats (3 months old) Induction: OVX-induced osteoporosis Treatment: 50 mg/kg of naringenin for 4 weeks Control: Negative: no treatment Positive: 0.2 mg/kg of estradiol oestrogen for 4 weeks	Trabecular width at epiphysis and metaphysis	Organic substances in tibia Periosteal transverse growth of the diaphysis Ratio of the transverse cross-sectional area of the marrow cavity/ diaphysis	Load, displacement, fracture load, and young modulus in femoral diaphysis Maximal load in femoral neck Displacement, maximal load fracture load, displacement for fracture load and Calcium and phosphorus content in femur, tibia, and L4 vertebrae young modulus in tibia metaphysis Transverse cross-sectional area of cortical bone in diaphysis and width of cartilage	Naringenin is safe for the skeleton and may have marginal skeletal beneficial effects on the bone.
[32]	Animal: 100 Y59 Female rats (3 months old, 200–250 g) Induction: Retinoic acid-induced bone loss Treatment: 100 mg/kg of naringenin for 14 days Control: Negative: no treatment Positive: 40 mg/kg of alendronate for 14 days	Calcium and phosphorus content in femur BMD in proximal and distal femur Femur length Glutathione content in liver and kidney	MDA in kidney and liver		Naringenin protects against retinoic acid-induced bone loss via antioxidant effects.
[23]	Animal: 15 Balb/cByJ mice Induction: OVX-induced osteoporosis Treatment: Preliminary studies—1 and 5 mg/kg/day of naringenin (25 μL) for 3 days in newborn mice Preventive studies—5 mg/kg/day of naringenin for 5 weeks in OVX mice Therapeutic studies—5 mg/kg/day of naringenin (i.p. injection) for 6 weeks in OVX mice Control: Negative: no treatment Positive: Preventive studies—17β estradiol (5 μg/kg/day for 5 weeks Therapeutic studies—40 μg/kg/day of human 1-34PTH (i.p. injection) for 6 weeks	Expression of Erα, Erβ, and BMP-2 in calvaria from newborn mice. BV/TV, Tb.N, CD, Tb.Sp, Tb.pf, and SMI at femur epiphysis in the preliminary study Expression of Runx-2 and type I collagen from bone marrow cells harvested from animals MAR and BFR/BS in preventive and therapeutic studies		BV/TV, Tb.N, Tb.Th, Tb.Sp, Tb.pf, and SMI in tibia trabecular preventive studies (Naringenin-6-*C*-glucoside could reverse these changes) BV/TV, Tb. N, Tb. Sp, SMI, and CD in distal femoral epiphysis and tibial proximal metaphysis (Naringenin-6-*C*-glucoside could reverse these changes)	Naringenin is less effective than Naringenin-6-*C*-glucoside in preventing bone loss.
[25]	Animals: 20 *C*-57/BL6 mice (8 weeks old Induction: Ti-particle-induced osteolysis. Treatment: Naringenin (10 mg/kg and 25 mg/kg) for 2 weeks Control: Negative: no treatment Positive: n.a	BV/TV in the ROI of the calvaria	Number of pores and percent porosity in the ROI of the calvaria TRAP-positive multinucleated osteoclasts		Naringenin prevents titanium particle-induced osteolysis caused by excessive osteoclast formation and activity.
[33]	Animal: 40 male Sprague-Dawley rats (46–54 g, 21-day-old) Induction: Soft diet-induced periodontal hypofunction. Treatment: 0.09%, 0.18%, 0.36%, and 0.72% of naringenin for 42 days Control: Negative: no treatment Positive: n.a		CAB-CEJ distance in buccal maxilla and mandible and lingua maxilla and mandible		Naringenin decreases the molar CAB-CEJ distance during alveolar development in young male rats.
[27]	Animal: Female ICR mice (4 weeks old, 23–29 g) Induction: OVX-induced osteoporosis Treatment: 3 and 10 mg/mL of naringenin every 2 days for 4 weeks Control: Negative: no treatment Positive: 17β-estradiol (0.1 mg/mL every 2 days for 4 weeks)	BMD and BMC ALP activity, BMP-2 expression in group treated with 10 mg/mL naringenin			Naringenin prevents bone loss due to ovariectomy.
[29]	Animal: 48 adult female adult Wistar rats (200–220 g) Induction: OVX-induced osteoporosis Treatment: 20 mg/kg naringenin for 60 days Control: Negative: no treatment Positive: 5.4 mg/kg raloxifene for 60 days	Serum ALP levels Improved cortical and trabecular bone architecture Well-organised bone matrix	Serum ACP levels		Oral naringenin nanosuspension prevents bone loss due to ovariectomy.
[34]	Animal: 50 Y59 Female rats (3 months old) Induction: Retinoic acid-induced bone loss Treatment: 100 mg/kg of naringenin for 14 days Control: Negative: no treatment Positive: 40 mg/kg of alendronate for 14 days		Serum β-CTx level Serum IL-1β, IL-6, TNFα and RANTES MDA in kidney and ovary GSH and CAT in kidney	Serum OCN level Calcium and phosphorus content in femur and serum BMD and BMC in proximal and distal femur Eroded endosteal bone surface, cortical bone thickness, and porosity	Naringenin prevents bone loss through antioxidant and anti-inflammatory effects.

Abbreviations: ↑, increase or upregulate; ↓, decrease or downregulate; ↔, no change; RANKL, receptor activator nuclear factor-B; M-CSF, macrophage colony-stimulating factor; IL-1α, -1β, -23, -8, -6, -4, -17, interleukin 1 alpha, 1 beta, -23, -8, -6, -4, -17; TNF-α, tumor necrosis factor alpha; OPG, osteoprotegerin; BMP-2, bone morphogenetic protein-2; Runx2,runt related transcription factor2; Osx, Osterix; p38MAPK, p38 mitogen activated protein kinase; Col1α2, collagen-type I-alpha 2; TRAP, tartrate-resistant acid phosphatase; sTRAP, secretory TRAP; cTRAP, cellular TRAP; mRNA, messenger ribonucleic acid; DC-STAMP, dendritic cell-specific transmembrane protein; NFATc1, nuclear factor of activated T-cell 1; V-ATPase, vacuolar type proton-translocating ATPase; NF-κB, nuclear factor-κB; ERK, extracellular signal-regulated kinase; JNK, *c*-Jun *N*-terminal kinase; IκBα, nuclear factor kappa light polypeptide gene enhancer B-cells inhibitor alpha; *p*-IκBα, phosphorylated nuclear factor kappa light polypeptide gene enhancer B-cells inhibitor alpha; BMSC, bone marrow-derived mesenchymal stem cells; hOB, human osteoblastic cells; pOB, primary osteoblastic cells; *c*-fos, proto-oncogene;CXCR4, chemokine receptor type 4; SDF-1a, stromal cell-derived factor; ALP, alkaline phosphatase; PI3K, phosphoinositide 3-kinase; Akt, protein kinase B; OVX, ovariectomy; BMD, bone mineral density; BMC, bone mineral content; OP, osteoporosis; MDA, malondialdehyde; ERα, estrogen receptor alpha; ERβ, estrogen receptor beta; BV/TV, bone volume/total volume; Tb.N, trabecular number; CD, connectivity density; Tb.sp, trabecular seperation; Tb.pf; SMI; Tb.Th., trabecular thickness; MAR, mineral apposition rate; BFR/BS, bone forming rate/bone surface; ROI, region of interest; CAB-CEJ, crestal alveolar bone-cemento-enamel junction; Ti, titanium; rBMSCs, rats bone marrow stroma cells; ACP, acid phosphatase; RANTES, regulated on Activation, Normal T Expressed and Secreted; β-CTx, β-CrossLaps; OCN, osteocalcin; GSH, reduced glutathione; CAT, catalase; hPDLSCs, human periodontal ligament stem cells.

## Data Availability

This manuscript does not contain original data.

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
