# Peer review of "A Scoping Review of the Skeletal Effects of Naringenin"

_nutrients, 2022, doi:10.3390/nu14224851_

Round 1

Reviewer 1 Report

Muhamad have conducted a systematic review of studies whose common objective was to evaluate the potential benefits of naringenin on bone. The authors indicate that they have included articles investigating the protective effects of naringenin with cells, animals or humans. The authors suggest that naringenin enhances osteoblastogenesis and bone formation and inhibits osteoclastogenesis and bone resorption. They conclude that Naringenin enhances bone formation while suppressing bone resorption, thus achieving its skeletal protective effects. They also recommend that it could be incorporated into the diet through fruit intake or supplements to prevent bone loss..

Major comments

The authors directly recommend the use of naringenin as a therapeutic resource to prevent bone loss. However, I have not been able to find a single randomized human clinical trial that proves its effectiveness in humans to prevent bone loss or as treatment for the osteoporosis. Nor have I found any randomized trials evaluating the effect of antioxidant principles in humans for the bone loss. The first reference in the paper [1] is an overview of the management of osteoporosis but no reference is made to naringenin. There is a review of the on Pharmacological and Analytical Aspects of Naringenin [2] but none of the included papers recommend it as a treatment for osteoporosis. All the papers evaluating the effects of naringenin included in this review are conducted in cells or animals, but there are none in humans. 

Human clinical trials may not have been conducted because preliminary studies have detected its low bioavailability. According to Shulman et al [3], naringenin suffers from low oral bioavailability critically limiting its clinical potential.

Could the authors clarify their recommendation for the use of naringenin and could it be an alternative or adjuvant treatment for osteoporosis? What would be the treatment guidelines? Why are there no randomized clinical trials in humans assessing its effectiveness despite the fact that there have been publications for quite some time assessing its effectiveness in animals?

This review includes work that is too disparate, mixing cell line and rat studies.  There is a recent meta-analysis [4] that includes only ovariectomized rats. The authors should have cited it.

Minor comments

1.     In the abstract, the authors say (sic) “Original research articles using cells, animals or humans to investigate the bone protective effects of 15 naringenin were included”. However, I have not been able to see any work with humans.

2.     Line 14 indicates that the systematic review was conducted in August 2022. I assume it is August 2021.

References.

[1] Sozen, T.; Ozisik, L.; Calik Basaran, N. An overview and management of osteoporosis. Eur. J. Rheumatol. 20174, 46–56, 373 doi:10.5152/eurjrheum.2016.048.

[2] Patel K, Singh GK, Patel DK. A Review on Pharmacological and Analytical Aspects of Naringenin. Chin J Integr Med. 2018 Jul;24(7):551-560. doi: 10.1007/s11655-014-1960-x. Epub 2014 Dec 10. PMID: 25501296.

[3] Shulman, M.; Cohen, M.; Soto-Gutierrez, A.; Yagi, H.; Wang, H.; Goldwasser, J.; Lee-Parsons, C.W.; Benny-Ratsaby, O.; Yar-513 mush, M.L.; Nahmias, Y. Enhancement of naringenin bioavailability by complexation with hydroxypropoyl-β-cyclodextrin. 514 PLoS One 20116, doi:10.1371/journal.pone.0018033. 

[4] Zhu Z, Xie W, Li Y, Zhu Z, Zhang W. Effect of Naringin Treatment on Postmenopausal Osteoporosis in Ovariectomized Rats: A Meta-Analysis and Systematic Review. Evid Based Complement Alternat Med. 2021 Feb 10;2021:6016874. doi: 10.1155/2021/6016874. PMID: 33628301; PMCID: PMC7889366.

Author Response

Dear reviewer, 

Thank you for reviewing our manuscript. We are grateful for the constructive comments and have responded to each of them in the response sheet attached. 

Thank you.

Reviewer 2 Report

In the present review article, entitled “A scoping review of the skeletal effects of naringenin” by Nor Muhamad et al., the authors used the keywords (1) naringenin AND (2) (bone OR osteoporosis OR osteoblasts OR osteoclasts OR osteocytes) to include 210 unique articles, which were from PubMed and Scopus and screen them by their rules to obtain 16 articles that met all the criteria. They integrated these 16 articles including in vitro and in vivo experiments, and sorted out the possible efficacy and mechanism of naringenin for osteoporosis. Overall, this review article is well done and worthy of publication in this journal. However, there are still few issues with this review article, as follows:

1.      Can the authors draw the structure of naringenin using computer software (such as ChemDraw) instead of using other sources?

2.      I doubt that there are more safety-related assessments reported for naringenin, including findings from in vitro and in vivo studies. If so, please add a little more information about the safety of naringenin.

3.      The authors mentioned that they have compiled the results of 16 articles, including 11 in vitro studies and 8 in vivo studies, as certain articles performed both in vitro and in vivo studies. Citation numbers 22, 24, 26, and 29 are the articles with both of in vitro and in vivo experiments. However, if you take a careful calculation, and deduct the number of in vitro and in vivo studies that are repeated, there are only 15 papers in total. In this regard, the authors are asked to confirm again rigorously.

4.      The authors changed the way of citing references after section 3.3, replacing the original citation number with the author's last name and year, which makes it difficult for readers to find the literature mentioned by the authors. In this regard, the authors are requested to have consistent citation way.

Author Response

(The authors gave the same response as above.)
